# Dynamics of Pd Subsurface Hydride Formation and Their Impact on the Selectivity Control for Selective Butadiene Hydrogenation Reaction

**DOI:** 10.3390/nano13061099

**Published:** 2023-03-19

**Authors:** Esther Asedegbega-Nieto, Ana Iglesias-Juez, Marco Di Michiel, Marcos Fernandez-Garcia, Inmaculada Rodriguez-Ramos, Antonio Guerrero-Ruiz

**Affiliations:** 1Dpto. Química Inorgánica y Técnica, Facultad de Ciencias, UNED, Av. de Esparta s/n, 28232 Las Rozas, Madrid, Spain; 2Instituto de Catálisis y Petroleoquímica, CSIC, c/Marie Curie No. 2, Cantoblanco, 28049 Madrid, Spain; 3ESRF—The European Synchrotron, 71 Avenue des Martyrs, 38000 Grenoble, France

**Keywords:** palladium catalysts, subsurface hydride, butadiene, partial hydrogenation, HEXRD

## Abstract

Structure-sensitive catalyzed reactions can be influenced by a number of parameters. So far, it has been established that the formation of Pd-C species is responsible for the behavior of Pd nanoparticles employed as catalysts in a butadiene partial hydrogenation reaction. In this study, we introduce some experimental evidence indicating that subsurface Pd hydride species are governing the reactivity of this reaction. In particular, we detect that the extent of formation/decomposition of PdHx species is very sensitive to the Pd nanoparticle aggregate dimensions, and this finally controls the selectivity in this process. The main and direct methodology applied to determine this reaction mechanism step is time-resolved high-energy X-ray diffraction (HEXRD).

## 1. Introduction

When studying heterogeneous catalysts, we encounter the concept of “structure sensitive reactions”. This term was first introduced by M. Boudart in the 1970s [1]. Briefly, for supported metal nanoparticles, some catalytic parameters, either specific activities for a given surface center (for example, turnover frequency) and/or the obtained selectivity values, vary drastically with the particle size.

The exact definition of catalytic surface sites is unclear due to the fact that, for instance, sometimes subsurface species from the bulk crystallites can participate in the reaction mechanism. For example, it has been reported that monoatomic dispersed Pd atoms over Cu surfaces constitute an excellent catalyst for the hydrogenation of acetylene to ethylene [2]. On the other hand, in the definition of a catalytic surface site, a discrete number (greater than one) of surface atoms can participate. These are the so-called ensemble structures or ensemble size effect, where the probability to find surface sites with the right size in terms of adjacent active centers governs the reaction rate [3]. In the surface site, a special arrangement of the metallic sites is considered to be involved in the definition of catalytic center. For example, it is assumed that B5-type surface sites, which consist of the arrangement of a layer of three atoms and another directly above of two atoms at the monoatomic step on a Ru (0001) terrace, are responsible for the activity of ruthenium catalysts [4]. Furthermore, J. A. Somorjai carried out studies where the relationship between reaction kinetics and atomic surface structure was questioned. This author deduced that structure sensitivity and insensitivity could be explained by the dynamic restructuring of metal surface induced by the presence of adsorbates [5]. More recently, a review dealing with the mechanistic and atomic-level insights into semihydrogenation reactions of alkynes and alkadienes was published [6]. In this sense, these authors classify under three categories, at the atomic level, the regulations for the active sites: site isolation, local environment regulation, and oxygen vacancy and interfacial sites.

Particularly, the industrially relevant studies of the selective hydrogenation of triple bonds/or conjugated double bonds in hydrocarbons are often catalyzed by Pd nanocrystallites. Although this noble metal is known to show the highest activities and selectivities in these reactions [7,8], to date, the mechanism behind its behavior has not been established. In general, this group of partial hydrogenation reactions has been classified as structure sensitive [9], and in earlier studies [10], the differences in catalytic selectivity have been associated with the strength of chemisorption of reactant or product molecules. According to a study carried out by D. Teschner et al. [11], the formation of the Pd/C surface phase was directly linked to the nature of the reactive molecule. These authors who studied the hydrogenation of alkynes stated that the formed Pd/C species alter the surface chemistry by decreasing the amount of activated subsurface hydrogen, thereby controlling the reactivity of the Pd catalyst. In another study, involving the hydrogenation of 1,3-butadiene on Pd (110)-oriented surfaces, it was observed that the presence of both weakly π-bonded 1,3-butadiene and subsurface hydrogen atoms is required to achieve catalytic activity [12]. Furthermore, Wu et al. [13] also attributes enhancements in catalytic performance to subsurface chemistry. In this cited work, focused on the partial hydrogenation of acetylene over Pd catalysts, the authors claim that the main reason for the catalytic performance of the active site can be tuned based on the distinctive electronic and geometric structures of Pd derived as a result of the presence of the corresponding subsurface heteroatoms.

In this contribution, for the first time, we demonstrate how the participation of hydride species formed inside Pd crystallites can control the activity and selectivity and establish the most favorable conditions for optimum results. We show how the nature and response characteristics of PdHx species are metallic particle size-dependent. Further, we present an operando study of the reversible transformation of Pd to PdHx during the hydrogenation reaction of 1,3-butadiene.

## 2. Materials and Methods

### 2.1. Synthesis of Materials

All catalysts studied in this work are composed of carbon-based materials as support and Pd nanoparticles as active phase. The former is either commercial high-surface-area graphite (G) purchased from Timcal (SBET = 296 m^2^/g) or lab-prepared graphenic materials. The graphenic materials were synthesized starting from commercial natural graphite obtained from Alfa Aesar, which is oxidized to produce the corresponding graphite oxide via the modified Brodie method, as has been detailed in our previous work [14]. This can be briefly described as follows: 200 mL of fuming nitric acid was introduced in a round-bottom flask. Thereafter, 10 g of natural graphite (grain size of 100 mesh) was added. The mixture is kept at 0 °C and under agitation. Thus, 80 g of KClO_3_ is slowly added over 1 h, after which the reaction is maintained at the abovementioned temperature for 21 h. Once completed, the obtained sample is filtered and washed until neutral pH is observed and then dried overnight in a vacuum furnace at 60 °C.

The graphite oxide (GO) prepared in the above paragraph is employed for the synthesis of two graphenic materials: GOE and GONE. GOE stands for exfoliated graphene oxide and is obtained by introducing the graphite oxide in a furnace and heating until 500 °C under inert atmosphere. As for GONE, which denotes N-doped exfoliated graphene oxide, this support is prepared by a solid-state synthesis method employing urea as the N precursor. For this, urea and GO are intimately mixed and ground, introduced in a furnace, and heated until 450 °C under nitrogen flow. Further details are specified in a previous work [15]. The three mentioned supports (G, GOE, and GONE) were impregnated with an ethanol–aqueous (1:1) solution of tetraamine palladium chloride. Quantities and concentrations were adjusted in order to obtain metal loadings of 1 and 2%. The prepared catalysts were dried overnight at 110 °C.

### 2.2. Characterization of Samples by TEM

Transmission Electronic Microscopy (TEM) was used to estimate the average particle size of Pd crystallites employing the formula d_Pd_ = ∑n_i_d_i_/∑n_i_. For this purpose, measurements were performed on a JEOL JEM-2100 field-emission gun electron microscope operating at 200 kV.

### 2.3. Catalytic Tests

Time-resolved high-energy X-ray diffraction (and mass spectrometry (MS) were the techniques employed, at the ID15A beamline of the European Synchrotron (ESRF), to monitor the whole process, which consisted of two main stages. Firstly, temperature-programmed reduction (TPR) is carried out on the catalyst in order to study the reducibility and formation of Pd species in the process of heating under a hydrogen atmosphere until 450 °C and cooling down to room temperature. This stage is necessary prior to the catalytic test as Pd has to be in its reduced state. The reductive gas is composed of 15% hydrogen in helium and a total flow rate of 20 mL/min. Secondly, the catalytic test consisting of the partial hydrogenation of butadiene was performed. For this, samples were pelleted in a size range of 0.075 to 0.150 mm and introduced in a 2 mm ID quartz capillary tube, which served as a fixed-bed continuous flow reactor. Glass wool stoppers are inserted at the top and bottom of the catalyst bed. Total flow rate is the same as for the TPR process, although in this case, butadiene (BD) is introduced in the reaction feed, maintaining a ratio to hydrogen of 1:5.

As for the HEXRD acquisition parameters, two sets of conditions were employed. On one hand, a wavelength of 0.177 Å (70 keV) was used and samples were placed at a distance of 65 cm from the Perkin Elmer area detector where data acquisition time was 30 s. The second set of experiments, named fast, on the other hand, were carried out at a distance of 105 cm from the detector, and the registered wavelength was 0.237 Å (52.33 keV) with exposure time reduced to 1 s. Both sets of experiments include the initial TPR and its consequent cooling to room temperature, and the reaction study, which consists of the introduction of the reaction mixture for 30 min at a constant temperature (switch 1) followed by a switch (switch 2) to reductive mixture (hydrogen diluted in helium) and thereafter increasing the temperature for the next reaction test at a higher temperature. This process is repeated for a number of different temperatures. The exhaust products from the reactor were analyzed using a European Spectrometry ecoSyst-P Man-Portable mass spectrometer with capillary inlet and heated inlet tubes. This serves to estimate the catalytic conversion of the reaction as it registers the characteristic mass fragments of the reactant, butadiene (*m*/*z* = 39), as well as the possible products, butene (partially hydrogenated, *m*/*z* = 41) and butane (completely hydrogenated, *m*/*z* = 43).

## 3. Results and Discussion

As expected, upon TPR, a diffraction peak at q~2.8 Å^−1^ identified as Pd (111) reflection demonstrates the presence of reduced Pd in all studied samples. Similar values have been reported elsewhere for this noble metal [16,17]. Figure 1 depicts the behavior for sample 1PdGOE. All other diffractogram sets are available in the Appendix A.

The evolution of PdHx species (diffraction peak at q~2.7 Å^−1^) formed during the cooling step, under a hydrogen atmosphere, is quite different in each case and is intimately linked with the Pd particle sizes. In summary, samples of smallest nanoparticle sizes (1PdG) apparently had no PdHx phase formation (see Appendix A), which is in agreement with the literature, where it has been said that below 2.5 nm, no hydride is formed [18]. In the case of the catalyst with largest particle sizes (1PdGOE), there was an almost complete transformation of Pd into PdHx on cooling the catalyst (Figure 1). This observation is consistent with a recent report, where large quantities of hydrogen absorbed on Pd and formation of palladium hydride are observed, at room temperature and under ambient pressure [19]. Surprisingly enough, the formed PdHx was consumed when put in contact with the reaction mixture (total flow 20 mL/min composed of H_2_ and butadiene (ratio 5:1) diluted in helium) at room temperature, and an evolution to metallic Pd occurred during the reaction (see Appendix A). This could explain the difference in conversion and selectivity values observed when comparing sample 1PdG (where no PdHx was formed) and 1PDGOE, as reported in Table 1.

In general, for Pd supported on graphenic or graphitic materials, an increase in the average Pd crystallite sizes, ranging from 3 to 10 nm, provokes a higher selectivity to butenes (from 27% to 92%). Hence, selectivity towards butenes is intimately linked with PdHx species present in this sample (1PdGOE). This tendency would be in agreement with studies carried out by Bauer et al. [20] where smaller Pd nanoparticles (1–2 nm) offered complete hydrogenation of propyne to propane at high conversion values, while high selectivity towards the partially hydrogenated product propene was favored with larger nanoparticles (about 15 nm). In this referenced study, the presence of subsurface C seemed to play a key role in the product distribution orientation. Yet, at that stage, transformation of PdHx to Pd was not considered or discussed as we are introducing in the present research. Using Pd supported on activated carbon catalysts, working in the hydrogenation of acetylene at 100 °C, both carbide and hydride were detected [21]. In another study by the same authors, XANES was said to discriminate between both phases [22]. Furthermore, Liu et al. [23] followed the formation of subsurface carbon on carbon-supported Pd catalysts (employing XAS and in situ XRD) at different temperatures for the partial hydrogenation of acetylene. It was concluded that the presence of this subsurface carbon improved the selectivity of the catalyst towards the unsaturated product, ethylene. The reason behind this would be that the formation of this species (Pd-carbide) hinders the formation of Pd hydride species and weakens the ethylene adsorption that gives rise to further hydrogenation into the saturated hydrocarbon. This was particularly relevant for reaction temperatures above 100 °C and especially at 250 °C, where the highest selectivity values towards ethylene were obtained. Hence, this behavior was said to be enhanced on increasing the reaction temperature where the penetration of carbon into Pd was more likely to occur. Nevertheless, at lower temperatures, such as 50 °C, as is in our case, the formation of the Pd-carbide phase is reduced, and the formation of the Pd-hydride is promoted. In summary, under our experimental conditions, the formation of Pd carbide cannot be excluded based only on the HEXRD results. Nevertheless, we could discard it for two main reasons. On one hand, the presence of this phase would yield methane, which was not detected among the products formed. Additionally, taking into consideration the low reaction temperature and the stoichiometric conversion of the peak into a metallic contribution during the hydrogenation process, the presence of PdHx is confirmed.

In order to have a deeper insight into the kinetic transformation of Pd to PdHx (necessary for the optimum performance of catalysts in this reaction) and vice versa, further experiments were performed. In this second set of experiments, some adjustments were made. On the one hand, and so as not to exclude the effect of support, graphite was still studied, although in this case, with higher Pd loading (2%) in order to provide larger particle sizes and discriminate whether formation of PdHx is dependent only on particle size and independent of the support. Acquisition time was reduced to 1 s to increase time resolution and experiments were performed with incident X-ray irradiation wavelength of 0.237 Å and a larger sample detector distance in order to improve the peak resolution (as detailed in the Materials and Methods section).

Therefore, three samples were studied: 2PdG, 1PdGOE, and 1PdGONE. All three introduced different and surprising discoveries on the formation of the Pd species. For 1PdGOE (Figure 1), both Pd and PdHx are formed during TPR and cooling to room temperature, respectively. Relevantly, for the 1PdGONE sample, the shape of the diffraction peak at 2.7 Å^−1^ suggests the presence of more than one contribution of PdHx, as can be viewed in Figure 2 (see also Appendix A where both (1PdGOE and 1PdGONE) are compared (Appendix A)). These two contributions correspond to two well-defined species: PdHx α (x ≤ 0.015) and PdHx β (0.5 ≤ x ≤ 0.75) [24]. This was most visible for 1PdGONE (Figure 2) but not for 2PdG, where only one contribution of Pd and PdHx species was observed (see also Appendix A). It has been reported that the miscibility gap between both phases (α and β) is narrowed with decreasing particle size [24]. Hence, for 2PdG, there is just one phase, possibly α-hydride, because it is well established that α-PdHx is preferentially formed over the smaller Pd crystallites (see comparative average particle sizes for these two samples, 1PdGONE and 2PdG, in Table 1). As can also be seen in Figure 2, the temporal evolution of the two Pd hydride peaks is different, indicating a difference in stability as a function of temperature.

The effect of hydrogen concentration, pressure, and temperature was revealed on the formation of Pd-hydride species (alpha and beta) [25,26]. These parameters govern the adsorption (physical and chemical) and diffusion processes involved in the formation of Pd-H species. The behavior of Pd thin films was compared with that of Pd NPs. As expected, lattice contraction is higher in NP with respect to bulk. This would justify the slightly lower contraction observed for 2PdG in comparison to the two others (1PdGOE and 1PdGONE), as it has been described elsewhere that lattice contractions increase with decreasing particle size for systems such as Au [27]. At the same time, in previous referenced work [25], on decreasing the temperature, a transition from alpha to beta is observed. Thus, we can also observe this in the cooling step of our TPR experiment for 1PDGONE (see Appendix A).

The transformation of PdHx to Pd during the hydrogenation reaction of butadiene is illustrated in Figure 3. When introducing the reaction mixture at a constant temperature of 30 °C, the Pd peak grows at the expense of the PdHx peak. Nonetheless, Figure 3 reveals that the transformation is not complete. On the other hand, PdHx is recovered after switching from the reaction mixture back to H_2_. The rate of transformation was also increased with increasing reaction temperature (30–70 °C) (see Appendix A). This reversible transformation was estimated to be about 80% (achieved after 4 min into the reaction) in the case of 1PdGOE in the whole range of reaction temperatures. Figure 4 and the Appendix A offer a more detailed scheme of this transformation for all the samples.

As for the 1PdGONE sample, subsequent switching to a reaction mixture (Bd + H2) and switching back to H_2_ again also leads to reversible transformation of PdHx into Pd, although to a lesser extent (Appendix A). In this case, the transformation achieved was about 66% after 4 min in the reaction (Figure 4).

As can be observed in Figure 4 (view also Appendix A), on switching from H_2_ atmosphere to reaction mixture, both phases of PdHx are not equally reduced. The β phase is preferentially and rapidly consumed. At this point, it can be deduced that the hydrogen content of the hydride is a determining factor for the optimal performance of Pd nanoparticles. The results presented in Figure 4 prove that the participation of the PdHx α phase is lower for 1PdGOE than for 1PdGONE and its concentration is inversely proportional to the selectivity towards butenes (Table 1).

Once more, the catalyst 2PdG exhibited the most extreme behavior. Firstly, at the lowest reaction temperature, 30 °C, PdHx transformation into Pd is negligible. At an intermediate temperature, 50 °C, this transformation is much slower than for the other two catalysts under the same conditions, reaching only about 20% throughout the reaction (Figure 4). At a higher temperature, 70 °C, although the transformation rate was still lower than with the other catalysts, a 100% reversible transformation after 20 min was reached (Appendix A). These results are once more related to the particle size of the Pd catalysts, which, as have been summarized in Table 1, follow the order: 1PdGOE > 1PdGONE > 2PdG.

At the same time, the evolution of the reaction products was followed by a mass spectrometer, so both butadiene conversion and selectivity towards butane and butenes were simultaneously analyzed with the Pd evolution. The results (Table 1) prove that the characteristics of the Pd phase (see Appendix A) and its evolution when switching reactants from H_2_ to the reaction C_4_H_6_/H_2_ mixture depend on the carbon support, which defines Pd nanoparticle size and, thus, the exposed PdHx phase and the active reaction temperature.

1PdGOE exhibits low conversion values, which increase slightly with increasing temperature. Referring to selectivity, the highest values towards butenes were observed with this sample, leaving butane production lower than 10% in all circumstances. In the case of 1PdGONE, the conversion was also low, although higher than that of 1PdGOE. It was also selective towards butenes, while butane production increased with increasing reaction temperatures. 2PdG gave high conversion values and low selectivity towards butenes, proving that butane formation is favored when small Pd nanoparticles are obtained, as no beta hydride phase is formed [28].

Therefore, time-resolved analysis of the combined in situ HEXRD and MS data shows that the reversible formation decomposition of the PdHx, which is dependent on particle size, is intimately linked with the conversion and the selectivity for a hydrogenation reaction. We conclude that the β-hydride phase is the species responsible for the production of butenes, while also suppressing the butane formation.

Taking into account the importance of the formed PdHx and its reversible transformation, it was necessary to take a deeper look into the subject. For this, a kinetic study at different reaction temperatures of the PdHx transformation during the hydrogenation reaction was performed. This was carried out employing the Avrami model, which is very suitable for phase transformations [29] (see Appendix A for data and analysis description). Table 2 summarizes the obtained transformation rate constants, k, associated with the apparent activation energy, while *n* is related to the critical rate-limiting step [30]. The value of k was highest for the sample of largest particle size 1PdGOE and lowest for that of the smallest, 2PdG. The exponential growth index, *n*, is particularly high for 2PdG. It is accepted that values of *n* > 4 are found in homogeneous transient nucleation, while heterogeneous nucleation gives exponent values <4 [31]. As transformation progresses, *n* becomes lower. The degree of transformation of 2PdG at 50 °C was only about 20%, hence, its high *n*. Values of about 1.5 can be ascribed to diffusion-controlled growth in pre-existing nuclei [32].

## 4. Conclusions

Depending on the Pd particle size, the reversible transformation of PdHx to Pd is more or less rapid. For instance, in the case of 1PdGOE, the transformation of the Pd hydride (having high β PdHx proportion) occurs with the highest velocity in comparison with the others. At the same time, kinetic studies prove that the hydride transformation of 2PdG proceeds at a slower rate. Relevantly, butadiene hydrogenation reaction behavior (activity and selectivity) as well as reversible Pd to PdHx transformation are intimately linked with the Pd nanoparticle size. Decreases in Pd particle size, increases in specific catalytic activity, and decreases in butene selectivity follow the same pattern: 1PdGOE > 1PdGONE > 2PdG. Larger Pd nanoparticles favor H solubility, reaching H-richer hydride species such as beta-hydride. Higher H content accelerated the transformation of PdHx into Pd during the reaction and, consecutively, the partial hydrogenation of butadiene to butenes. On the other hand, Pd and alfa-PdH lead to the butane production.

In this study, by simultaneously exploring the formation of subsurface Pd-hydride and the catalytic properties (selectivity and activity in the butadiene hydrogenation), it is demonstrated that the participation of hydride species formed inside Pd crystallites is controlling the dynamics of the hydrogen activation and insertion (reversible transformation of Pd to PdHx) in the reaction products.

## Figures and Tables

**Figure 1 nanomaterials-13-01099-f001:**
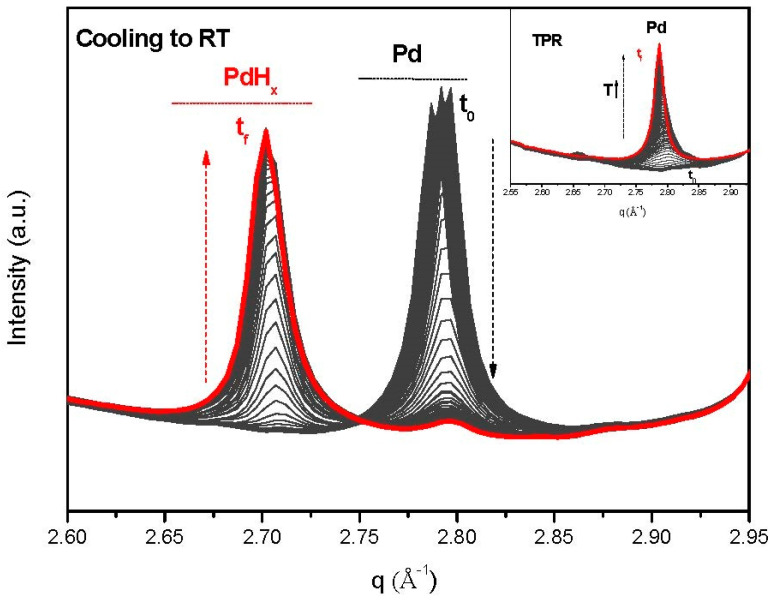
Formation of metallic Pd during TPR process for 1PdGOE (insert) and formation of PdHx upon cooling. Black initial (t_0_) and red final (t_f_) patterns.

**Figure 2 nanomaterials-13-01099-f002:**
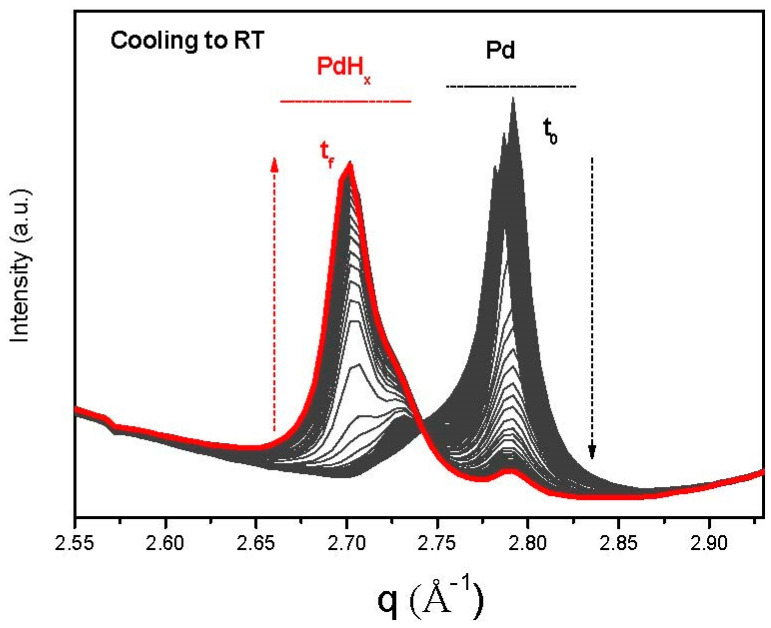
Transformation of Pd to PdHx during cooling to RT for 1PdGONE. Black initial (t_0_), red final (t_f_).

**Figure 3 nanomaterials-13-01099-f003:**
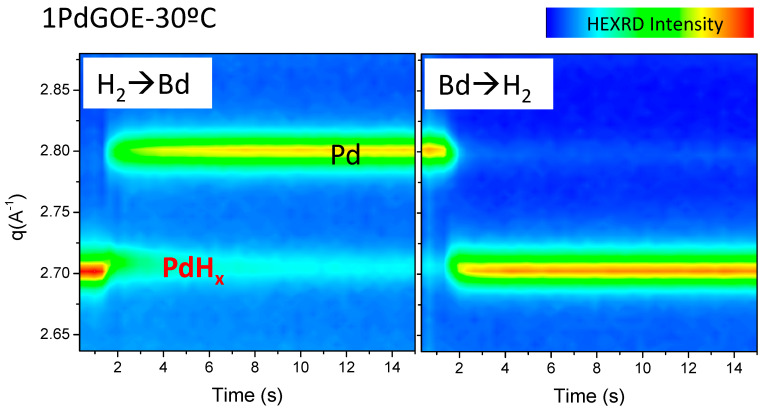
Transformation of PdHx to Pd during butadiene hydrogenation at 30 °C for 1PdGOE. Figure based on data detailed in Appendix A section.

**Figure 4 nanomaterials-13-01099-f004:**
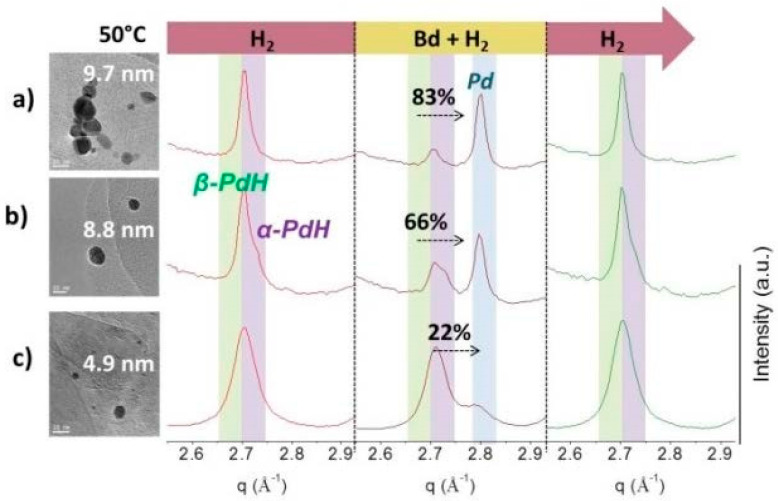
Final steady-state HEXRD patterns (selected Pd(111) region) after H_2_ exposure at 50 °C, subsequent switching to reaction mixture (Bd + H2), and back switching to H_2_ again for Pd samples: (**a**) 1PdGOE, (**b**) 1PdGONE, and (**c**) 2PdG.

**Table 1 nanomaterials-13-01099-t001:** Particle size of the Pd nanoparticles and steady-state conversions and selectivities at 50 °C reaction temperature.

Samples	Particle Size nm (TEM)	Conversion (%)	Selectivity to Butenes (%)
1PdG	3.2	89	27
2PdG	4.9	89	27
1PdGONE	8.9	15	88
1PdGOE	9.7	8.9	92

**Table 2 nanomaterials-13-01099-t002:** Kinetic parameters of the metal to hydride conversion.

Samples	Particle Size nm (TEM)	Ln k	*n*
2PdG	4.9	−25.2	8.6
1PdGONE	8.9	−3.5	1.9
1PdGOE	9.7	−2.7	1.6

## Data Availability

Not applicable.

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
