# Peer review of "Dynamics of Pd Subsurface Hydride Formation and Their Impact on the Selectivity Control for Selective Butadiene Hydrogenation Reaction"

_nanomaterials, 2023, doi:10.3390/nano13061099_

Round 1

Reviewer 1 Report

The authors indicate that sub-surface Pd hydride species are governing the reactivity of butadiene hydrogenation reaction using the time-resolved high-energy X-ray diffraction (HEXRD). They detect that the extent of formation/decomposition of PdHx species is sensitive to the Pd nanoparticle aggregate dimensions, and that finally controls the selectivity of the reaction. This work is very interesting. This paper can be accepted.

Author Response

The authors indicate that sub-surface Pd hydride species are governing the reactivity of butadiene hydrogenation reaction using the time-resolved high-energy X-ray diffraction (HEXRD). They detect that the extent of formation/decomposition of PdHx species is sensitive to the Pd nanoparticle aggregate dimensions, and that finally controls the selectivity of the reaction. This work is very interesting. This paper can be accepted.

Answer: The authors appreciate the reviewer's positive view of this contribution paper. We also want to highlight the good understanding that our manuscript has had.

Reviewer 2 Report

The manuscript report on the dynamics of Pd sub-surface hydride formation and on the selectivity control for selective 4 butadiene hydrogenation reaction.

The work it's interesting and maybe too specific. Experimental data are consistent with the results. Other similar works have been published but not for graphene oxide based-material. The manuscript can be accepted in current form but I suggest to add in the references the paper "Structure of a seeded palladium nanoparticle and its dynamics during the hydride phase transformation" Suzana et al Nature Communication  2021, 4, 64.

Reference n. 4, the Journal name should be in italic.

Author Response

The manuscript report on the dynamics of Pd sub-surface hydride formation and on the selectivity control for selective 4 butadiene hydrogenation reaction.

The work it's interesting and maybe too specific. Experimental data are consistent with the results. Other similar works have been published but not for graphene oxide based-material. The manuscript can be accepted in current form but I suggest to add in the references the paper "Structure of a seeded palladium nanoparticle and its dynamics during the hydride phase transformation" Suzana et al Nature Communication  2021, 4, 64.

Reference n. 4, the Journal name should be in italic.

Answer: The authors appreciate the reviewer's positive view of this contribution. Following the reviewer's recommendations, reference 4 has been corrected and a new reference incorporated into the revised text.

Actions: reference 4, Journal name, with italic in the revised manuscript. A new reference [19] incorporated and discussed in the revised manuscript.

Reviewer 3 Report

In this work, the author explains the relationships between Pd-HX species and Pd nanoparticle dimensions, as well as the particle size and butene selectivity by using the time-resolved high-energy X-ray diffraction and TEM techniques. The results are interesting, and the work worth to be published in the journal. However, the following issues should be addressed before its reconsideration.

1. The first is that the effect of support should be taken into consideration, because the support has an important influence on the particle size. In other words, why did the authors choose carbon-based material as support for palladium?

2. Could the palladium-carbide be converted into Pd-H?

3. The data of 2PdGONE and 2PdGOE should be added to prove the relationship between particle size and catalytic performances.

4. More data are better to be added in the kinetics study of Fig. S18 b, d, and f.

Author Response

In this work, the author explains the relationships between Pd-HX species and Pd nanoparticle dimensions, as well as the particle size and butene selectivity by using the time-resolved high-energy X-ray diffraction and TEM techniques. The results are interesting, and the work worth to be published in the journal. However, the following issues should be addressed before its reconsideration.

Answer: The authors appreciate the reviewer's critical view of this contribution paper. The interesting points raised by this reviewer have been in detail commented, point by point, below.

  1. The first is that the effect of support should be taken into consideration, because the support has an important influence on the particle size. In other words, why did the authors choose carbon-based material as support for palladium?

Answer: This is a very good question. We had been working with carbonaceous support materials for many years and we have expertise in preparing supported metal nanoparticles (in the present case Pd) with controlled particle sizes. In the present research carbon or graphitic materials offer an interesting possibility: model support (inert) to achieve well controlled Pd nanoparticles (sized dimensions). And of course these catalytic materials have been tested in the butadiene partial hydrogenation and have exhibited remarkable properties.   

  1. Could the palladium-carbide be converted into Pd-H?

Answer: Ours experiment of TPR indicate that initially the metallic Pd nanoparticles are converted to hydride with decreasing the temperature down to (or near) room temperature. Once these nanoparticles are in contact with the reaction mixture metallic Pd is progressively regenerated. These to reactions operated reversibly. As we studied the process under mild reaction conditions, we consider that carbide formation, a part of non-detected should be rather difficult. However, we agree with the reviewer that carbide species are relevant intermediated for many reactions occurring at higher reaction temperatures.  

  1. The data of 2PdGONE and 2PdGOE should be added to prove the relationship between particle size and catalytic performances.

Answer: Unfortunately the experiments described in this manuscript have been conducted at the synchrotron ESRF installations. This mean that we are unable to perform additional experiments for many years. But we agree with the reviewer that more data should be very illustrative to demonstrate the reaction mechanism hear proposed for this reaction and this type of catalysts.

  1. More data are better to be added in the kinetics study of Fig. S18 b, d, and f.

Answer: The same comment that for the point 3 can be argued.